# Game Space and Game Situation as Mediators of the External Load in the Tasks of School Handball

**DOI:** 10.3390/ijerph20010400

**Published:** 2022-12-27

**Authors:** Sebastián Feu, Juan Manuel García-Ceberino, María Gracia Gamero, Sergio González-Espinosa, Antonio Antúnez

**Affiliations:** 1Universidad de Extremadura, Facultad de Educación y Psicología, Avenida de Elvas s/n, 06006 Badajoz, Spain; 2Optimization of Training and Sports Performance Research Group (GOERD), University of Extremadura, 10003 Cáceres, Spain; 3Faculty of Education, Psychology and Sports Science, University of Huelva, 21007 Huelva, Spain; 4Faculty of Health Sciences, International University of La Rioja, 26006 Logroño, Spain; 5Faculty of Sports Science, University of Extremadura, 10003 Cáceres, Spain

**Keywords:** planning, external load, service teachers, SIATE

## Abstract

The teacher’s didactic intervention also requires knowledge and control of learning tasks’ workloads. The objectives of the study were as follows: (i) to quantify the subjective external load-eTL of tasks framed in didactic units designed by in-service teachers; and (ii) to analyze the differences in the subjective eTL according to the game situation and the game space. A total of 306 learning tasks designed by seven in-service teachers (five men and two women), with more than 10 years of teaching practice, were analyzed. These tasks were coded through the Integral System for Training Task Analysis (SIATE, acronym in Spanish). The interobserver reliability of the coded variables obtained a considerable concordance (MKfree > 0.70). The results indicated that there were significant differences in the subjective eTL according to the game situation and game space. The situations of small-sided games in numerical equality or inequality and full games, in medium spaces or large spaces, presented a higher subjective eTL and therefore the highest physiological and motor demands on students. The inclusion of attacking or defending players and an adequate selection of the game space indicated the importance of planning and organizing learning tasks.

## 1. Introduction

The Didactics of Physical Education and school sports pursue the transmission of knowledge and practical lessons in an orderly manner through the didactic intervention of the teacher. This intervention refers to the actions of the teacher during the teaching-learning process, such as the choice of teaching styles, the design of activities or tasks, and the selection of resources that are necessary for motor practice [1].

The teaching of sport in physical education classes has been carried out from the repetition of exercises aimed at the acquisition of certain technical movements [2]. This traditional (or technical) method takes teaching out of context, since these technical gestures are practiced in isolation from real game situations to be incorporated into the game later. This fact suggests a lack of motivation and that little time is devoted to the development of decision-making [3,4]. The resulting low learning outcomes lead to low satisfaction among students [5]. Because of the drawbacks of the technical conception, various studies recommend the application of the comprehensive (or tactical) methods in primary education [2,6,7]. The comprehensive methods consider the student as the center of the teaching−learning process, where tactical components and decision-making are predominant. Students build their own learning and increase their amusement, because they practice sports in situations similar to real games [8]. These methods use modified games, called small-sided games (SSGs), which are characterized by using adaptations of the sport in its original version: dimensions of the game space, the number of participants, rules of the game, and so on, always considering the characteristics and learning of the students [9].

Within the actions of the teacher, this study is oriented to the design of the learning tasks framed in didactic units, coinciding with the pre-service phase (prior to teaching) defined by Viciana [10]. In this regard, learning tasks constitute the most specific unit of sports programming [11], and they are aimed to achieve the proposed didactic objectives through the practice of the curricular contents and to increase the motivation and interest of the students [12]. Their design and organization should not be the product of the teacher’s invention, improvisation, and creativity, but rather the result of a thoughtful and thorough process [13].

In the design of motor tasks, the teacher must consider the heterogeneity of physical education classes, as well as design tasks that meet the needs of students. In this way, it will be possible to achieve the desired learning outcomes while maintaining the interest of students in the tasks used [14]. Likewise, with the purpose that disconnected objectives are not set in the didactic units and that students retain the learning acquired, it is necessary that teachers use an adequate number of sessions (>10 sessions) [15]. In addition to cognitive implications, the tasks have an external load (eTL) assumed by the student [16]. The eTL is a physical demand that the students face during sports practice [17]. Teachers can manipulate the eTL imposed by tasks by modifying the structural and formal parameters of the tasks, such as the game space, the number of subjects involved, the duration of the tasks, the counting of points, and/or the encouragement of the teacher [18]. These constraints, together with feedback (quantity and quality), teacher expectations, teacher-student communication, communication between students, support for autonomy, or class size, influence the level of learning [15]. However, teachers tend to plan the sessions of didactic units based on their experiences, unaware of the eTL to which the students are exposed [19].

The teacher must have knowledge about the sport to be taught and identify variables or constraints that define tasks [20]. The Integral System for Training Task Analysis (SIATE) [21] makes it possible to record and analyze different factors that affect the teaching−learning process of invasion sports: (i) pedagogical variables (they offer information on the characteristics of the tasks); (ii) organizational variables (they offer information on the organizational aspects of the school group, the temporal distribution of the task, and the organization of material resources); and (iii) external load variables (allow obtaining the quantification of the subjective eTL caused by the teaching tasks) [21]. The use of this instrument helps teachers to plan more precisely and to optimize the teaching−learning process [22].

In the scientific literature, there are different studies that have analyzed the subjective eTL caused by learning tasks designed by teachers (in training) to teach invasion sports in their physical education classes: in handball [23], in soccer [24,25], or in basketball [26,27]. Likewise, there are studies that have analyzed the subjective eTL of tasks framed in basketball teaching programs [28] and soccer [29] in the primary level. These studies on the quantification of the subjective eTL accumulated in physical education classes show teachers guidelines on how to design tasks to reach adequate levels of eTL (physical demand), similar to the real demands of the sport practiced, and focus on the teaching of invasion sports under comprehensive methods.

Therefore, the objectives of this study were as follows: (i) to quantify the subjective eTL caused by tasks framed in handball didactic units designed by in-service teachers; and (ii) to analyze the differences in the subjective eTL, depending on the situation and the game space.

## 2. Materials and Methods

A comparative and cross-sectional associative study was carried out [30], aimed at quantifying and discovering the association between the eTL and the situation and the game space in the tasks of six handball teaching units.

### 2.1. Sample

The analysis of the eTL was carried out with 306 tasks designed by seven teachers in the service phase (5 men and 2 women), with more than 10 years of teaching experience. They taught in state schools in Spain. The tasks were designed for handball teaching units applied in the fifth and sixth grades of primary education. A convenience sampling was carried out, selecting those in-service teachers who wanted to participate in the research and who had greater ease of access.

### 2.2. Instrument

The tasks framed in the didactic units were categorized through the task analysis system called SIATE [21]. The analysis of the eTL carried out using SIATE yields valid and reliable data, being used in various research projects [23,28,29]. A high correlation was found between the eTL (subjective) measured by SIATE, the eTL (objective) recorded with inertial devices using the Player Load variable, and the internal load measured through the heart rate [31,32].

### 2.3. Procedure

Firstly, the in-service teachers were asked for a didactic unit planned and used by themselves for teaching handball in educational centers where they currently carry out their teaching work. They previously signed a written informed consent form.

Subsequently, after collecting their didactic units, the tasks were coded with the selected variables with the help of the SIATE instrument [21]. The coder’s background was as follows: (i) Doctor; (ii) university lecturer in invasion sports and/or sports teaching methods; (iii) level III coach in the invasion sport of handball; and (iv) publications on the teaching of invasion sports, specifically handball. Together with two other observers, interobserver reliability was calculated using the Free-Marginal Multirater Kappa (Multirater Kfree) program [33], analyzing 84 tasks that represented more than 27.40% of the total sample. The percentage of tasks was higher than that used in a similar study [26] for the calculation of interobserver reliability. Thus, almost perfect reliability was obtained for the variables game situation (MKfree = 0.88) and degree of opposition (MKfree = 0.83), considerable reliability for the variables density of the task (MKfree = 0.78) space (MKfree = 0.75), percentage of simultaneous participants (MKfree = 0.70), competitive load (MKfree = 0.74), and cognitive involvement (MKfree = 0.70) [34,35].

The SIATE instrument was used. It is a methodological system for recording and analyzing different factors that affect the teaching of invasion sports. These factors fall into three main groups of variables: pedagogical, organizational, and load. The quantification of the eTL caused by the tasks was calculated using six load variables (categorical-ordinal) recorded in this instrument. This quantification was obtained by adding the value assigned within each of the six variables (from 1 to 5 points), with a range between 6 and 30 points. Four ranges are established to categorize its value: 6–12 (very low level), 13–18 (medium−low level), 19–24 (medium−high level), and 25–30 (very high level) [21]. Likewise, in order to establish comparisons between groups, the variables “game situation” and “game space” were used. Both variables were structured with a definition of five categories. Table 1 shows the variables studied and their category structures.

The study protocol respected the ethical guidelines of the Helsinki Declaration of 1975 (with modifications in subsequent years) and the Organic Law 3/2018 of 5 December, on the protection of personal data and guarantee of digital rights (BOE, 294, 6 December 2018), to guarantee the ethical considerations of scientific research with human subjects. In addition, the approval of the Bioethics Committee of the University was obtained (approval number: 105/2022, June 29).

### 2.4. Statistical Analysis

A descriptive statistical analysis was performed for all qualitative (*n* and %) and quantitative (M and SD) variables in the study. The association between categorical variables was estimated through the chi-square coefficient (X^2^) and Cramer’s V coefficient (Vc) [37]. In addition, Fisher’s exact test (f) was used with the Monte Carlo method, as very low frequencies were found [38]. The degree of association between variables was estimated from the four ranges of association defined by Crewson [39]: small (values < 0.100), low (values between 0.100 and 0.299), moderate (values between 0.300 and 0.499), and high (values > 0.500). The degree of association between the variable categories was interpreted through the adjusted standardized residuals (ASRs) (>|1.96|) of the contingency tables [38].

Finally, a descriptive and inferential analysis of the eTL was carried out, depending on the situation and the game space. After verifying that the assumption of normality of the data was not met (*p* < 0.05), it was decided to use the non-parametric Kruskal−Wallis H test. For these analyses, the statistical program SPSS 25 (IBM Corp. Released 2017. IBM SPSS Statistics for Windows, Version 25, IBM Corp, Armonk, NY, USA) was used.

## 3. Results

Differences in the level of the eTL were found, depending on the game situation (X^2^ = 449.419, *p* < 0.001; f = 392.545, *p* < 0.001; Vc = 0.700, *p* < 0.001) and the game space (X^2^ = 228.372, *p* < 0.001; f = 211.357, *p* < 0.001; Vc = 0.499, *p* < 0.001). Table 2 and Table 3 show the descriptive results and the ASR of the task-induced eTL, depending on the situation and the game space, respectively.

An inferential analysis of the eTL was performed in the animation (warm-up) and the fundamental (content-specific and higher intensity activities) parts (*n* = 255 tasks), depending on the situation and the game space. The tasks of cooldown were excluded from this analysis due to their low eTLs, as in most cases they did not work on the objectives of the session and they interfered with the knowledge of the eTL of the tasks aimed at learning the specific contents of handball. Significant differences were obtained, depending on the game situation (X^2^ = 208.406; gl = 4; *p* < 0.001; d = 4.235) (Figure 1). SSGs in equality and full games elicited the greatest eTL.

Table 4 shows the pairwise comparisons (for the eTL) of the game situation categories. Statistically significant differences were observed between all groups, except between SSGs in inequality, 1 × 1 groups, and SSGs in equality (adjusted *p* > 0.05).

Depending on the game space where the tasks were developed, significant differences were also obtained (X^2^ = 82,339; gl = 4; *p* < 0.001; d = 1.351) (Figure 2). Thus, large spaces and repetition of large spaces elicited the greatest eTL.

Table 5 shows the pairwise comparisons (for the eTL) of the game space categories. Statistically significant differences were observed between all groups, except between static activities and small spaces (adjusted *p* > 0.05).

## 4. Discussion

The use of the load variables in the design of tasks is essential for optimizing learning [23], because physical education sessions must involve both cognitive and physical components. The aim of this study was to quantify the eTL caused by tasks framed in didactic units designed by in-service teachers for the teaching of handball in the school environment and to analyze the differences in the eTL, depending on the situation and the game space. The results obtained indicate statistically significant differences in the design of the eTL regarding the situation and the game space.

The in-service teachers analyzed used a low number of sessions (between seven and nine sessions). According to Hébrard [40], 10 sessions are insufficient to achieve the desired level of learning in physical education and for students to retain learning over time. Some studies consider that 8 h, under the supervision of a professional, would be an adequate number to produce significant improvements in students with a comprehensive methodology [4,5]. In their annual program, teachers plan a large amount of content, but without going too deeply into it, due to lack of time, since in Spain only 2 or 3 h of physical education are taught per week per group/class [15].

Regarding the game situation, the ASR indicated that there were more cases than expected in the ranges of the higher eTL in the categories of full games and SSGs of numerical equality. The numerical equality tasks involve a higher eTL than the tasks posing superiorities or inferiorities, since they represent real situations of the competition itself [41]. Similarly, numerical equality tasks (1 × 1) and numerical inequality of SSGs presented a medium−high eTL level, while unopposed tasks presented a medium−low eTL level. Low levels of eTL are associated with a greater use of simple application exercises [29], which are characterized by using situations without opponents/defenders. The predominance of unopposed tasks is characteristic of a traditional or technical method [28,42].

In terms of the game space, there were more cases than expected at higher eTL levels in the large space and large space repetitions categories. Likewise, the categories of static activities and small spaces presented medium and medium−low levels. Therefore, the greater the distance covered, the higher the subjective eTL level of the task. However, Barbero [43] states that covering a greater number of meters does not necessarily mean that the participant uses his maximum aerobic potential, despite the fact that there is a high correlation.

In this study, it has been possible to verify that the game situation, that is, the presence and relationship between the number of attacking and defending players, and the repeated practice of sports games in large spaces favored a higher subjective eTL. In other studies, it has been observed that the modification of the variables that make up the subjective eTL (OD + TD + SP + CL +GSp + CI) also had a direct effect on the objective eTL (kinematic and neuromuscular), measured with inertial devices, and on the internal load of the task [31], being necessary to take these parameters into account when designing learning tasks. On the other hand, in physical education classes, the use of teaching methods centered on the student and on the understanding of the game, based on played situations, leads to higher levels of the eTL, measured subjectively with the SIATE [28,29], and the objective eTL measured through inertial devices [44] than tasks designed under a traditional methodology. All these results highlight the importance of the selection of teaching methods and the planning of learning tasks.

The external load borne by students is conditioned by organizational variables and the game situation [45] as well as by the pedagogical model teaching [46]. It is necessary for teachers to know the structural and formal parameters of tasks, so that they can rigorously design and sequence the learning tasks [47].

## 5. Conclusions

This research invites us to reflect on the way in which teachers plan their handball teaching sessions at school. In addition, in the search for the optimization of the teaching-learning process of invasion sports, it is essential to make right decisions in the planning of variables such as the playing space and the number of players present in the learning tasks, especially for effects that they provoke in the external load that learners bear. These variables must be considered in order to adapt and improve the physical development of students.

## Figures and Tables

**Figure 1 ijerph-20-00400-f001:**
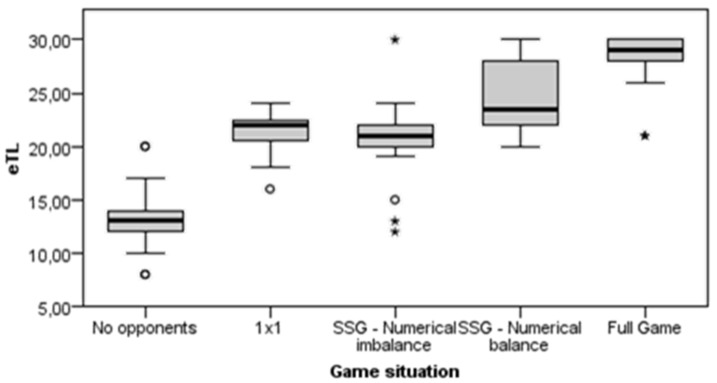
Results of the differences in the eTL according to the game situation. Note: the circle represents the outliers, while the asterisk represents the extreme values.

**Figure 2 ijerph-20-00400-f002:**
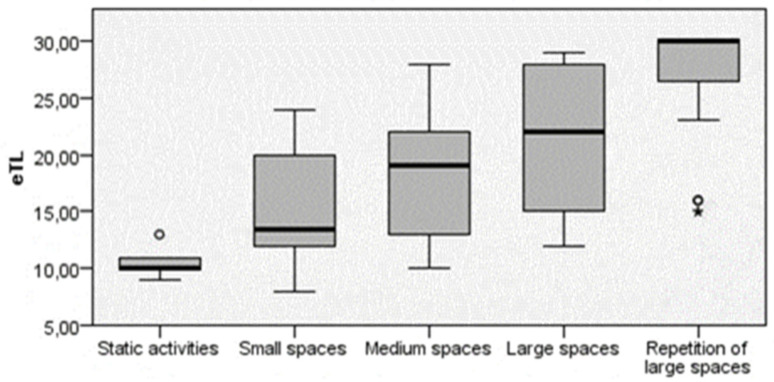
Results of the differences in the eTL according to the game space. Note: the circle represents the outliers, while the asterisk represents the extreme values.

**Table 1 ijerph-20-00400-t001:** Independent and dependent variables and their category structures.

Variables		Categories
Independent variables	Game situation (GS)	No opposition (1 × 0, 2 × 0, 3 × 0,…)
		Numerical equality (1 × 1)
		SSGs numerical inequality (2 × 1, 3 × 1, 3 × 2,…)
		SSGs numerical equality (2 × 2, 3 × 3, and 4 × 4)
		Full games (5 × 5, 6 × 6, 7 × 7, and N × N)
	Game space (GSp)	Static activities
		Small spaces (1/4 field)
		Medium spaces (1/2 field)
		Large spaces (the entire field)
		Repetition in large spaces
Dependent variables	Load variables	Opposition degree (OD)
		Task density (TD)
		% Simultaneous performers (SP)
		Competitive load (CL)
		Game space (GSp)
		Cognitive involvement (CI)
eTL quantification was calculated as: OD + TD+ SP + CL + GSp + CI.		

Note: SSGs, small-sided games; N × N, The teacher divides the class into two teams or similar. We indicated SSGs when referring to reduced situations and full games when referring to more realistic game situations [36].

**Table 2 ijerph-20-00400-t002:** Descriptive results and ASRs of the task-induced eTL, depending on the game situation.

		eTL Levels				
		Low	Low−Medium	Medium−High	High	Total
No opposition	n	72	73	2	0	147
	% within the GS	49.0	49.7	1.4	0.0	100.0
	% within the eTL level	94.7	86.9	2.2	0.0	48.0
	% total	23.5	23.9	0.7	0.0	48.0
	ASR	9.4 *	8.4 *	−10.4 *	−8.0 *	
1 × 1	n	0	3	16	0	19
	% within the GS	0.0	15.8	84.2	0.0	100.0
	% within the eTL level	0.0	3.6	17.8	0.0	6.2
	% total	0.0	1.0	5.2	0.0	6.2
	ASR	−2.6 *	−1.2	5.4 *	−2.1 *	
SSGs in numerical inequality	n	3	7	56	1	67
	% within the GS	4.5	10.4	83.6	1.5	100.0
	% within the eTL level	3.9	8.3	62.2	1.8	21.9
	% total	1.0	2.3	18.3	0.3	21.9
	ASR	−4.4 *	−3.5 *	11.0 *	−4.0 *	
SSGs in numerical equality	n	0	0	14	12	26
	% within the GS	0.0	0.0	53.8	46.2	100.0
	% within the eTL level	0.0	0.0	15.6	21.4	8.5
	% total	0.0	0.0	4.6	3.9	8.5
	ASR	−3.1 *	−3.3 *	2.9 *	3.8 *	
Full games	n	1	1	2	43	47
	% within the GS	2.1	2.1	4.3	91.5	100.0
	% within the eTL level	1.3	1.2	2.2	76.8	15.4
	% total	0.3	0.3	0.7	14.1	15.4
	ASR	−3.9 *	−4.2 *	−4.1 *	14.1 *	
Total	n	76	84	90	56	306
	% within the GS	24.8	27.5	29.4	18.3	100.0
	% within the eTL level	100.0	100.0	100.0	100.0	100.0
	% total	24.8	27.5	29.4	18.3	100.0

Note: GS, game situation; eTL, external load; SSGs, small-sided games. * ASR > |1.96|.

**Table 3 ijerph-20-00400-t003:** Descriptive results and ASRs of the task-induced eTL, depending on the game space.

		eTL levels				
		Very Low	Low	High	Very High	Total
Static activities	n	45	4	0	0	49
	% within the GSp	91.8	8.2	0.0	0.0	100.0
	% within the eTL level	59.2	4.8	0.0	0.0	16.0
	% total	14.7	1.3	0.0	0.0	16.0
	ASR	11.8 *	−3.3 *	−4.9 *	−3.6 *	
Small spaces	n	21	15	20	0	56
	% within the GSp	37.5	26.8	35.7	0.0	100.0
	% within the eTL level	27.6	17.9	22.2	0.0	18.3
	% total	6.9	4.9	6.5	0.0	18.3
	ASR	2.4 *	−0.1	1.1	−3.9 *	
Medium spaces	n	9	44	41	19	113
	% within the GSp	8.0	38.9	36.3	16.8	100.0
	% within the eTL level	11.8	52.4	45.6	33.9	36.9
	% total	2.9	14.4	13.4	6.2	36.9
	ASR	−5.2 *	3.4 *	2.0 *	−0.5	
Large spaces	n	1	17	27	19	64
	% within the GSp	1.6	26.6	42.2	29.7	100.0
	% within the eTL level	1.3	20.2	30.0	33.9	20.9
	% total	0.3	5.6	8.8	6.2	20.9
	ASR	−4.8 *	−0.2	2.5 *	2.6 *	
Repetition in large spaces	n	0	4	2	18	24
	% within the GSp	0.0	16.7	8.3	75.0	100.0
	% within the eTL level	0.0	4.8	2.2	32.1	7.8
	% total	0.0	1.3	0.7	5.9	7.8
	ASR	−2.9 *	−1.2	−2.4 *	7.5 *	
Total	n	76	84	90	56	306
	% within the GSp	24.8	27.5	29.4	18.3	100.0
	% within the eTL level	100.0	100.0	100.0	100.0	100.0
	% total	24.8	27.5	29.4	18.3	100.0

Note: GSp, game space; eTL, external load. * ASR > |1.96|.

**Table 4 ijerph-20-00400-t004:** Pairwise comparisons (for the eTL) of the game situation categories.

Category 1 × Category 2	Test Statistic	Std. Error	Std. Test Statistic	Sig.	Adjusted Sig.
No opponents—SSGs in inequality	−91.783	11.790	−7.785	0.000 *	0.000 *
No opponents—1 × 1	−107.273	18.296	−5.863	0.000 *	0.000 *
No opponents—SSGs in equality	−138.418	16.069	−8.614	0.000 *	0.000 *
No opponents—full games	−170.644	13.382	−12.752	0.000 *	0.000 *
SSGs in inequality—1 × 1	15.491	19.308	0.802	0.422 *	1.000 *
SSGs in inequality—SSGs in equality	−46.635	17.212	−2.705	0.007 *	0.067 *
SSGs in inequality—full games	−78.861	14.735	−5.352	0.000 *	0.000 *
1 × 1—SSGs in equality	−31.145	22.181	−1.404	0.160 *	1.000 *
1 × 1—full games	−63.371	20.319	−3.119	0.002 *	0.018 *
SSGs in equality—full games	−32.226	18.339	−1.757	0.079 *	0.789 *

Note: SSGs, small-sided games. * *p* < 0.05.

**Table 5 ijerph-20-00400-t005:** Pairwise comparisons (for the eTL) of the game space categories.

Category 1 × Category 2	Test Statistic	Std. Error	Std. Test Statistic	Sig.	Adjusted Sig.
Static activities—small spaces	−61.854	34.354	−1.800	0.072 *	0.718 *
Static activities—medium spaces	−102.623	33.605	−3.054	0.002 *	0.023 *
Static activities—large spaces	−140.666	34.166	−4.117	0.000 *	0.000 *
Static activities—RLS	−200.088	36.128	−5.538	0.000 *	0.000 *
Small spaces—medium spaces	−40.769	12.211	−3.339	0.001 *	0.008 *
Small spaces—large spaces	−78.812	13.679	−5.761	0.000 *	0.000 *
Small spaces—RLS	−138.234	18.029	−7.667	0.000 *	0.000 *
Medium spaces—large spaces	−38.043	11.671	−3.260	0.001 *	0.011 *
Medium spaces—RLS	−97.465	16.557	−5.887	0.000 *	0.000 *
Large spaces—RLS	−59.421	17.668	−3.363	0.001 *	0.008 *

Note: RLS, repetition of displacements in large spaces. * *p* < 0.05.

## Data Availability

Data will be available upon reasonable request from the corresponding author.

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
