# Peer review of "Game Space and Game Situation as Mediators of the External Load in the Tasks of School Handball"

_ijerph, 2022, doi:10.3390/ijerph20010400_

Round 1

Reviewer 1 Report

Are some suggestions that can help better structure the scientific text:

11The first sentence of the “abstract” must be deleted. Does not fit the text.

22.  Start the “abstract” with a brief introduction, before exposing the objectives.

33. In the “abstract” and “introduction”, add the meaning of the acronym eTL (the first time it appears).

44. At the end of the “introduction”, in the first objective, add the name of the sport (handball).

55.  Materials and Methods:

• Describe the primary objective as described at the end of the introduction.

• Describe whether there was approval by the Ethics Committee.

• Describe whether there was a informed consent from the participants.

• Sample: Better characterize the sample (number, age of participants, gender, beyond schooling...); Describe if there were inclusion and exclusion criteria; Describe where the participants come from and selection method.

• Instrument: Describe better what it is and how the SIAT.

66. Results:

• Identify in the text, the parts that refer to “figures 1 and 2”.

• Improve the quality of figures 1 and 2.

Author Response

We would like to express our gratitude to reviewer 1 for the time in reviewing our manuscript and for providing us comments helpful to improve this manuscript quality. We have answered their concerns (all corrections were marked in red and/or change control).

--------------------

Reviewer’ note: The first sentence of the “abstract” must be deleted. Does not fit the text.

Authors’ response: We agree with you. The first sentence of the abstract has been deleted.

--------------------

Reviewer’ note: Start the “abstract” with a brief introduction, before exposing the objectives.

Authors’ response: Thank you very much for your suggestion. An introductory sentence has been added (line 15 to 16).

--------------------

Reviewer’ note: In the “abstract” and “introduction”, add the meaning of the acronym eTL (the first time it appears)

Authors’ response: Thank you very much for your suggestion. The meaning of the acronym eTL (external load) has been added the first time it appears.

--------------------

Reviewer’ note: At the end of the “introduction”, in the first objective, add the name of the sport (handball)

Authors’ response: We agree with you. The name of the sport (handball) has been added to the first objective (line 95).

--------------------

Reviewer’ note: Describe the primary objective as described at the end of the introduction.

Authors’ response: The objective was described in the line 99 to 101.

--------------------

Reviewer’ note: Describe whether there was approval by the Ethics Committee.

Authors’ response: Based on your comment, the code of approval of the Bioethics Committee has been added (line 153 to 154).

--------------------

Reviewer’ note: Describe whether there was an informed consent from the participants.

Authors’ response: Thank you very much for your suggestion. It has been indicated that written informed consent was requested from the participants (line 118).

--------------------

Reviewer’ note: Sample: Better characterize the sample (number, age of participants, gender, beyond schooling...); Describe if there were inclusion and exclusion criteria; Describe where the participants come from and selection method.

Authors’ response: Thank you very much for your comment. It has been indicated that the sample selection was made using convenience sampling (line 106 to 107).

--------------------

Reviewer’ note: Instrument: Describe better what it is and how the SIATE.

Authors’ response: Based on your comment, the SIATE instrument has been better described (line 132 to 134).

--------------------

Reviewer’ note: Identify in the text, the parts that refer to “figures 1 and 2”.

Authors’ response: Thank you very much for your comment. The parts that refer to "Figures 1 and 2" have been identified in the text (lines 188 and 199).

--------------------

Reviewer’ note: Improve the quality of figures 1 and 2.

Authors’ response: We agree with your comment, but Figures 1 and 2 have been exported from the SPSS program.

Reviewer 2 Report

In this study, the authors study the external training load for different tasks in handball. In particular, they investigate the differences dependent on the game space or game situations. The authors conclude that there are significant difference between the external training load according to the specific details of the task in terms of game space or game situation.

In my opinion, there is lack of novelty in this study. It is quite straightforward that the external training load depends on the game space or game situation of a specific task that is considered. Although education classes are not frequently considered, there are many studies that investigate a wide range of load variables in sports. The analysis presented here is quite limited as the authors only use a single subjective measure and consider some specific pre-defined categories. Therefore, I am not convinced this work adds much to the existing scientific literature and I can not recommend  publication.

Some more specific comments:

1) Abstract: Remove first sentence.

2) The introduction is very comprehensive. As many side-paths are described in much detail, this section is sometimes hard to follow for the reader. Therefore, I would encourage the authors to make the introduction concise and to the point.

3) Line 123: Which didactic units?

4) Line 127 - 129: How did the authors test interobserver reliability? Did multiple observers code the same task? If so, how many observers? Moreover, why are only 84 tasks considered? 

5) The presentation of the results is not clear to me. In particular, what do the authors mean with "% within the GS", "% within the eTL level" and "% total"? Moreover, why is the total number of tasks in Table 3 equal to 306 and in Table 2 only 292?

6) Line 177 - 183: What are parts of the animation session and the fundamental parts? Also was is "return to calm"?

7) I guess the eTL are compared in Table 3 and 4? For completeness, please add information what is compared.

8) Line 192 - 193: Could the authors further explain this?

Author Response

We would like to express our gratitude to reviewer 2 for the time in reviewing our manuscript and for providing us comments helpful to improve this manuscript quality. We have answered their concerns (all corrections were marked in red and/or change control).

--------------------

Reviewer’ note: In my opinion, there is lack of novelty in this study. It is quite straightforward that the external training load depends on the game space or game situation of a specific task that is considered. Although education classes are not frequently considered, there are many studies that investigate a wide range of load variables in sports. The analysis presented here is quite limited as the authors only use a single subjective measure and consider some specific pre-defined categories. Therefore, I am not convinced this work adds much to the existing scientific literature and I can not recommend  publication.

Authors’ response: We disagree with your comment. The knowledge and use of the load variables in the design of the learning tasks is essential for optimizing learning, because Physical Education sessions must involve both the cognitive and the physical components. In this study, it has been possible to verify that the game situation, that is, the presence and relationship between the number of attacking and defending players, and the repeated practice of sports games in large spaces favored a higher subjective external load. It has been observed that the modification of the variables/constraints have a direct effect on the subjective external load. Therefore, it is necessary for teachers to know the structural and formal parameters of the tasks so that they can rigorously design and sequence the learning tasks.

In addition, the introduction sectiton has been modified and restructured.

--------------------

Reviewer’ note: Abstract: Remove first sentence.

Authors’ response: We agree with you. The first sentence of the abstract has been deleted.

--------------------

Reviewer’ note: Line 123: Which didactic units?

Authors’ response: Based on your question, we refer to didactic units designed by in-service teachers (line 119).

--------------------

Reviewer’ note: How did the authors test interobserver reliability? Did multiple observers code the same task? If so, how many observers? Moreover, why are only 84 tasks considered?

Authors’ response: Thank you very much for your questions. Based on the literature, more than 20% of the tasks (i.e., 84 tasks) representative of the total sample were analyzed for the calculation of interobserver reliability. In this case, the objective was to analyze the reliability of the principal observer (line 125 to 127).

--------------------

Reviewer’ note: The presentation of the results is not clear to me. In particular, what do the authors mean with "% within the GS", "% within the eTL level" and "% total"? Moreover, why is the total number of tasks in Table 3 equal to 306 and in Table 2 only 292?

Authors’ response: Thank you very much for your questions. There was a mistake in the Table 2, which has been corrected. These percentages (Tables 2 and 3) were obtained using contingency tables, which indicate the percentage from the perspective of the game space and game situation when crossed with the categories of eTL.

--------------------

Reviewer’ note: Line 177 - 183: What are parts of the animation session and the fundamental parts? Also was is "return to calm"?

Authors’ response: Thank you very much for your questions. These concepts have been specified (line 182 to 184).

--------------------

Reviewer’ note: I guess the eTL are compared in Table 3 and 4? For completeness, please add information what is compared.

Authors’ response: We agree with you. It has been specified that Tables 3 and 4 correspond to the eTL.

--------------------

Reviewer’ note: Line 192 - 193: Could the authors further explain this?

Authors’ response: Thank you very much for your question. The indicated lines have been explained.

Round 2

Reviewer 2 Report

First, I would like to thank the authors for taking into account my comments and revising their manuscript. Unfortunately, the authors could not convince me of the scientific relevance of this work. It is already known that the external training load depends on the characteristics of a training session, such as distance covered or number of high-intensity efforts. Since these measures are strongly related to the game space or game situation, it is rather straightforward that the external training load also depends on the game space and game situation. Moreover, there are already multiple studies that investigate the external training load in detail. Therefore, I believe this study does not contain enough novelty to recommend publication.

Author Response

We would like to express our gratitude again to reviewer 2 for the time in reviewing our manuscript.

--------------------

Reviewer’ note: It is already known that the external training load depends on the characteristics of a training session, such as distance covered or number of high-intensity efforts. Since these measures are strongly related to the game space or game situation, it is rather straightforward that the external training load also depends on the game space and game situation. Moreover, there are already multiple studies that investigate the external training load in detail. Therefore, I believe this study does not contain enough novelty to recommend publication.

Authors’ response: We disagree with you. This study coincides with the pre-service phase (prior to teaching) as defined by Viciana [10], a phase little studied in the scientific literature. Physical education teachers should be aware of the factors/constraints they can use to control the external load of the learning tasks framed in the didactic units (before the students play them). It contrasted the factors/constraints that seven in-service teachers used when planning the teaching of school handball, which has made it possible to observe deficiencies in such planning in accordance with current trends.

Viciana, J. Planificar en Educación Física (1ª Ed.); Inde: Barcelona, España, 2002.

--------------------

Reviewer’ note: The introduction is very comprehensive. As many side-paths are described in much detail, this section is sometimes hard to follow for the reader. Therefore, I would encourage the authors to make the introduction concise and to the point.

Authors’ response: Thank you very much for your comment. The introduction was restructured and summarized in the previous revision, with a final length of 870 words.

Kind regards.